# Providing normative information increases intentions to accept a COVID-19 vaccine

Alex Moehring [1,2], Avinash Collis [3,7], Kiran Garimella[4,7], M. Amin Rahimian [2,5,7], Sinan Aral [1,2,6] & Dean Eckles [1,2,6] ✉

Despite the availability of multiple safe vaccines, vaccine hesitancy may present a challenge to successful control of the COVID-19 pandemic. As with many human behaviors, people's vaccine acceptance may be affected by their beliefs about whether others will accept a vaccine (i.e., descriptive norms). However, information about these descriptive norms may have different effects depending on the actual descriptive norm, people's baseline beliefs, and the relative importance of conformity, social learning, and free-riding. Here, using a pre-registered, randomized experiment ($N = 484,239$) embedded in an international survey (23 countries), we show that accurate information about descriptive norms can increase intentions to accept a vaccine for COVID-19. We find mixed evidence that information on descriptive norms impacts mask wearing intentions and no statistically significant evidence that it impacts intentions to physically distance. The effects on vaccination intentions are largely consistent across the 23 included countries, but are concentrated among people who were otherwise uncertain about accepting a vaccine. Providing normative information in vaccine communications partially corrects individuals' underestimation of how many other people will accept a vaccine. These results suggest that presenting people with information about the widespread and growing acceptance of COVID-19 vaccines helps to increase vaccination intentions.

Nonpharmaceutical interventions in response to outbreaks of infectious disease, such as the COVID-19 pandemic, often depend on the behavioral responses of the public for their effectiveness. Even with the availability of vaccines, success depends on people's choices to accept, or even seek out, the vaccine[1], since even low vaccine refusal rates can prevent achieving herd immunity[2,3]. Given the significant ethical and practical challenges of imposing vaccine mandates[4–6], it is important to understand how public health messaging can increase acceptance of safe and effective COVID-19 vaccines. Many messaging strategies address individual barriers to vaccination, such as

complacency and inconvenience[7], as well as the perceived risk of both vaccines and the disease[1,8–10]. Early trials provide evidence that reminder messages can at least cause people to receive vaccines earlier[11].

It may be important to look beyond individuals to consider how public health messaging can also leverage the significant roles of social networks (broadly defined) in shaping individual vaccination decisions[12–16]. Rather than being a small factor, there is growing evidence that people's preventative health behaviors are dramatically influenced by many social and cultural factors, with implications for

[1]Sloan School of Management, Massachusetts Institute of Technology, Cambridge, MA, USA. [2]MIT Initiative on the Digital Economy, Massachusetts Institute of Technology, Cambridge, MA, USA. [3]McCombs School of Business, The University of Texas at Austin, Austin, TX, USA. [4]School of Communication and Information, Rutgers University, New Brunswick, NJ, USA. [5]Department of Industrial Engineering, University of Pittsburgh, Pittsburgh, PA, USA. [6]Institute for Data, Systems, and Society, Massachusetts Institute of Technology, Cambridge, MA, USA. [7]These authors contributed equally: Avinash Collis, Kiran Garimella, M. Amin Rahimian. ✉e-mail: eckles@mit.edu

COVID-19[17,18]. In the United States, for example, analyses of mobility data during the COVID-19 pandemic revealed that people's mobility behaviors vary with their partisan affiliation[19] and media consumption[20,21] and are affected by the behaviors of their social connections[22]. In particularly relevant work, Bicchieri et al.[23] find that experimental variations in descriptive and injunctive norms induce substantial variation in predictions about the individual's likelihood of engaging in preventative behaviors in various vignette scenarios.

Acceptance of COVID-19 vaccines likely involves substantial social influence, but theory is not entirely clear on whether learning how many others are accepting a vaccine will increase or decrease acceptance. Positive peer effects can arise due to information diffusion[24,25], conformity and injunctive norms[15,26], inferring vaccine safety and effectiveness from others' choices[27,28], or pro-social motivations such as altruism[29,30] and reciprocity[31]. On the other hand, negative effects of others' acceptance can arise as a result of free-riding on vaccine-generated herd immunity, even if only partial or local[32,33]. The empirical evidence on when positive peer effects[28,34,35] or free-riding[32] may dominate is inconclusive. Furthermore, the effects of incorporating accurate information about others' into messaging strategies will depend on what that information is, i.e., how prevalent is vaccine acceptance in a given reference group? In the presence of positive peer effects, we may generally wonder whether the true rate of vaccine uptake is high enough that emphasizing this information increases acceptance. Thus, we need further empirical guidance about scalable and effective messaging strategies leveraging social influence. That is, while some interpretations of the theoretical and empirical literature could motivate emphasizing high rates of vaccine acceptance in public health communications, little is known about how realistic interventions of using messages with factual information about others' vaccine acceptance will affect intentions to accept the COVID-19 vaccines.

Here, we provide evidence, from a large-scale randomized experiment embedded in an international survey, that accurate information about descriptive norms—what other people do, believe, or say—often has positive effects on intentions to accept new vaccines for COVID-19. Furthermore, we generally rule out large negative effects of

such information. We find mixed evidence that information on descriptive norms impacts mask wearing and no statistically significant evidence that it impacts physical distancing.

## Results

Through a collaboration with Facebook and Johns Hopkins University, and with input from experts at the World Health Organization and the Global Outbreak Alert and Response Network, we fielded a survey in 67 countries in their local languages, yielding over two million responses[36]. This survey assessed people's knowledge about COVID-19, beliefs about and use of preventative behaviors, beliefs about others' behaviors and beliefs, and economic experiences and expectations. Recruitment to this survey was via messages from Facebook to its users that encouraged potential respondents to help with research on COVID-19 (Supplementary Fig. S1). While it is often impossible to account for all factors that may jointly determine selection into the sample and survey responses, our collaboration with Facebook allows using state-of-the-art, privacy-preserving weighting for non-response using rich behavioral and demographic variables, as well as further weighting to target the adult population of each country[36,37]. All analyses presented here use these survey weights to ensure our results are as representative of these countries' adult populations as possible. Additional information about the weights, and the main analyses replicated without using weights, are in Supplementary Note 5.2.

### Trends in vaccination intentions and social norms

This survey has documented substantial variation in stated intentions to take a vaccine for COVID-19 when one is available to the respondent, with, for example, substantial changes over time and some countries having much larger fractions of people saying they will take a vaccine than others (Fig. 1). However, a plurality consistently say they will accept a vaccine and only a (often small) minority say they will refuse one. This is consistent with other smaller-scale national[10,38] and international[39] surveys. There is also substantial variation in what fraction of other people respondents think will accept the vaccine, and these beliefs often substantially differ from country-wide levels of

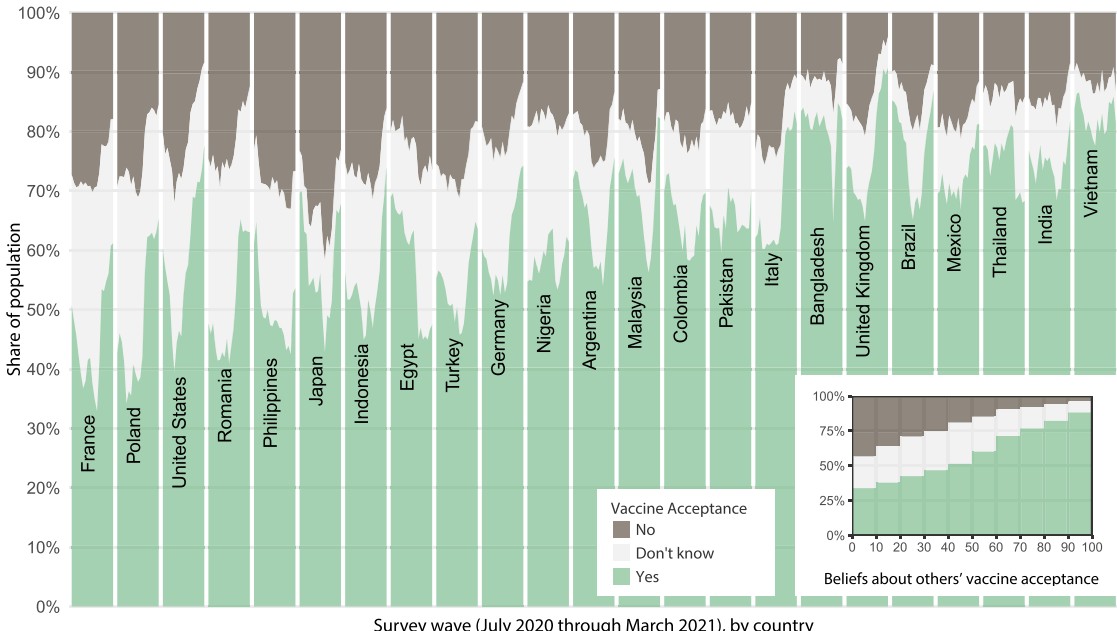

**Fig. 1 | Time series of COVID-19 vaccine acceptance from July 2020 to March 2021 by country.** Shown are the 23 countries with repeated data collection over time. "Yes" also includes respondents indicating they already received a vaccine. Within each country, there are 19 points representing a time-series across the 19 waves of the survey. (inset) Pooling data from all 23 countries, people who believe a larger fraction of their community will accept a vaccine are on average more likely to say they will accept a vaccine; this is also true within each included country (Supplementary Fig. S15). Source data are provided as a Source Data file.

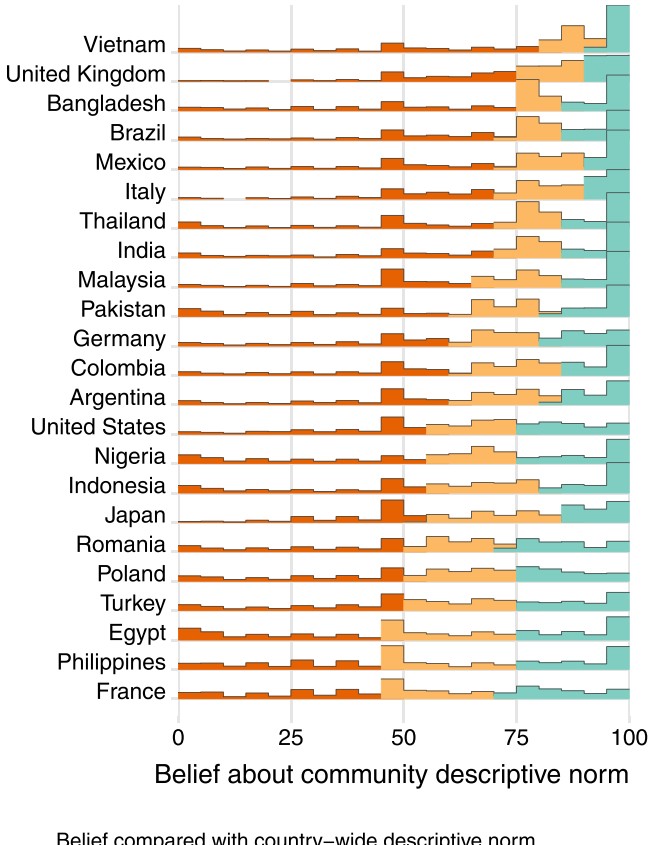

**Belief compared with country–wide descriptive norm**

■ below narrow %    ■ between narrow and broad %    ■ above broad %

**Fig. 2 | Within-country distributions of beliefs about descriptive norms.** Plot of within-country distributions of beliefs about descriptive norms ("Out of 100 people in your community, how many do you think would take a COVID-19 vaccine if it were made available?") during the experimental period (October 2020 to March 2021). To enable comparison with actual country-wide potential vaccine acceptance, these histograms are colored by whether they are below (red) the narrow ("Yes" only) definition of vaccine acceptance, between (yellow) the narrow and broad ("Yes" and "Don't know") definitions, or above (teal) the broad definition. Source data are provided as a Source Data file.

vaccine acceptance (Fig. 2). This deviation can have multiple causes, including responding with round numbers; but we posit this is at least partially because some people have incorrect beliefs about descriptive norms. Underestimation of vaccine acceptance by others could be partially caused by processes—such as news coverage of the challenges posed by vaccine hesitancy or diffusion of anti-vaccine messages on social media—that make hesitancy more salient. Beliefs about descriptive norms are in turn positively correlated with vaccine acceptance (Fig. 1, inset, Supplementary Fig. S15, and Supplementary Note 6), likely reflecting many processes, such as geographic and social clustering of vaccine hesitancy, but also causal effects of beliefs about others on intentions to accept a vaccine[36]. Public health communications could present information about norms, perhaps correcting some people's overestimation of the prevalence of vaccine hesitancy. Unlike other ongoing, frequently observable preventative behaviors, like mask wearing, people may have little information about whether others intend to or have accepted a vaccine—which suggests messages with this information could have substantial effects.

### Randomized experiment
To learn about the effects of providing normative information about new vaccines and other preventative health behaviors, beginning in October 2020, for the 23 countries with ongoing data collection in the

survey[36], we presented respondents with accurate information based on how previous respondents in their country had answered a survey question about vaccine acceptance, mask wearing, or physical distancing. We randomized at what point in the survey this information was presented, which behavior the information was about, and how we summarized previous respondents' answers—enabling us to estimate the effects of presenting information about descriptive norms on people's stated intentions to accept a vaccine.

In the case of vaccine acceptance, we told some respondents, "Your responses to this survey are helping researchers in your region and around the world understand how people are responding to COVID-19. For example, we estimate from survey responses in the previous month that X% of people in your country say they will take a vaccine if one is made available", where X is the (weighted) percent of respondents saying "Yes" to a vaccine acceptance question. Other respondents received information on how many "say they may take a vaccine", which is the (weighted) percent who chose "Yes" or "Do not know" for that same question. (The weighted estimate is preferred to the unweighted estimate and corresponds to the methods used elsewhere in, e.g., dashboards and reports on this survey[36].) We randomize whether this information occurs before or after a more detailed vaccine acceptance question and whether it uses the broad (combining "Yes" and "Do not know") or narrow ("Yes" only) definition of potential vaccine accepters, which allows us to estimate the causal effects of this normative information. (When the detailed vaccine acceptance question occurs after the normative information, it is always separated by at least one intervening screen with two questions, and it is often separated by several screens of questions.) Here, we focus on comparisons between providing the normative information about vaccines before or after measuring outcomes (e.g., vaccine acceptance); in the SI, we also report similar results when the control group consists of those who received information about other behaviors (i.e., about mask wearing and distancing), which can avoid concerns about differential attrition and researcher demand.

On average, presenting people with normative information on the share of respondents in a country who will accept a vaccine increases stated intentions to take a vaccine, with the broad and narrow treatments causing 0.039 and 0.033 increases on a five-point scale (95% confidence intervals: [0.028, 0.051] and [0.021, 0.044], respectively; Fig. 3 and Supplementary Note 5). For mask wearing and physical distancing, the effects are smaller and often not statistically distinguishable from zero. Focusing on vaccination intentions, the distribution of responses across treatments (Fig. 4a) reveals that the effects of the broad (narrow) treatment are concentrated in inducing an additional 1.6% (1.1%) of people to say they will at least "probably" accept the vaccine, and moving 1.9% (1.7%) to "definitely" (Supplementary Table S8). Note that these statements are about effects on the cumulative distribution of the vaccine acceptance scale (e.g., the proportion answering at least "Probably"). The proportion answering exactly "Probably" is similar across conditions (Fig. 4a), consistent with the treatment shifting some respondents from "Unsure" to "Probably" but also some from "Probably" to "Definitely". For the broad treatment, this represents a 4.9% relative reduction in the fraction of people choosing a response that is "unsure" or more negative, a 2.4% relative increase in the fraction choosing at least "Probably", and a 3.8% relative increase in the fraction of people choosing "Yes, definitely". A post hoc analysis also concluded that these effects are largest among people who answer "Don't know" to the baseline vaccine acceptance question (Fig. 4b and Supplementary Table S12), consistent with the idea of targeting vaccine fence-sitters[40]. As a comparison point, these effects are over a third of the size of the total increase in vaccine acceptance from November 2020 to January 2021 across all 23 countries (0.11 on the five-point scale)—a period that featured frequent and widely-distributed vaccine-related news (for this comparison, we restrict to the time period before vaccines were available to the public as this

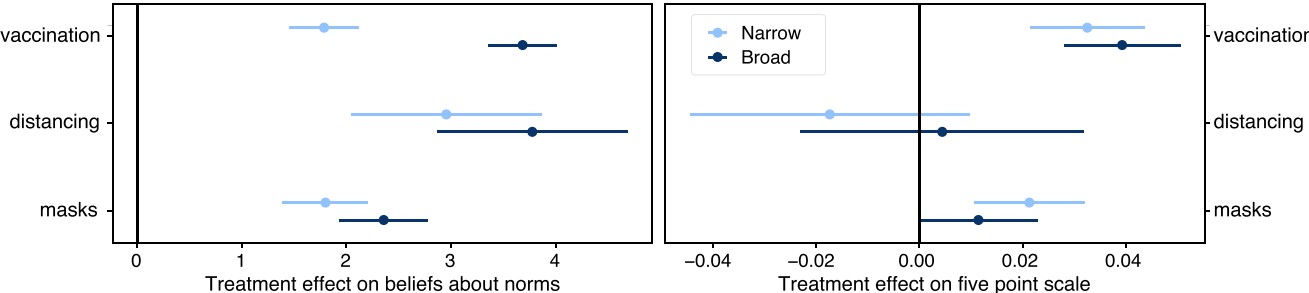

**Fig. 3 | Treatment effects on beliefs and intentions.** (left) Effect on beliefs about descriptive norms. Coefficients on treatment from a regression of beliefs about norms on treatment status, including centered covariates and interactions. In this analysis, treated respondents are those who receive the treatment before the question eliciting beliefs about norms. This will not agree, in general, with the treatment status for the main analysis given the randomized question order in the survey. There are $n = 304{,}840$ responses in the masking analysis, $n = 70{,}078$ in the physical distancing analysis, and $n = 356{,}004$ in the vaccination analysis. (right) Effect on intentions. Coefficients from regression of intentions on treatment, centered covariates, and their interactions. There are $n = 323{,}085$ responses in the masking analysis, $n = 85{,}619$ in the physical distancing analysis, and $n = 365{,}593$ in the vaccination analysis. Error bars are 95% confidence intervals centered around mean estimates. Source data are provided as a Source Data file.

question was only shown to those who had reported not having already received a vaccine).

These effects on vaccine acceptance can be at least partially attributed to changes in respondents' beliefs about these descriptive norms. We can examine this because the survey also measured respondents' beliefs about vaccine acceptance in their communities (as displayed in Fig. 2), and we randomized whether this was measured before or after providing the normative information. As expected, the normative information treatment increased the fraction of people that the respondents estimate will accept a vaccine (Fig. 3 and Supplementary Note 4). Among those respondents for whom we measured these normative beliefs prior to treatment, we can examine how treatment effects varied by this baseline belief. In particular, we classify respondents according to whether their baseline belief was above the broad ("may take") number, under the narrow ("will take") number, or between these two numbers. (The question measuring beliefs about descriptive norms asks about "your community", while the information provided is for the country. Thus, for an individual respondent, these need not match exactly to be consistent.)

Consistent with the hypothesis that this treatment works through revising beliefs about descriptive norms upwards, we find significant effects of the normative information treatment in the groups that may be underestimating vaccine acceptance—the under and between groups (Fig. 4b), though the smaller sample sizes here (since these analyses are only possible for a random subset of respondents) do not provide direct evidence that the effect in the under group is larger than that in the above group (difference in treatment effects of $0.021 \pm 0.024$ s.e., $P = 0.38$, and $0.023 \pm 0.022$ s.e., $P = 0.31$, for broad and narrow treatments, respectively). A post hoc analysis to address possible mismeasurement due to a preference to report round numbers (by removing those who reported they believe 0%, 50%, or 100% of people in their community would accept a vaccine) was likewise consistent with this hypothesis (difference in treatment effects of $0.057 \pm 0.027$ s.e., $P = 0.03$, and $0.027 \pm 0.026$ s.e., $P = 0.3$, for broad and narrow treatments, respectively). We had also hypothesized that the broad and narrow treatments would differ from each other in their effects on respondents in the between group, but we found no such evidence (difference in treatment effects of $0.004 \pm 0.021$ s.e., $P = 0.87$). In order to be accurate, these treatments also differed in their wording, which could have counteracted any effect of the difference in the numbers presented.

Having fielded this experiment in 23 countries, we can estimate and compare treatment effects internationally, while keeping in mind that estimates for individual countries have lower precision. Using a linear mixed-effects model, we estimate positive effects in the vast majority of countries (Fig. 4c). While estimates for some countries

are larger (e.g., Pakistan, Vietnam) and some are smaller (e.g., Nigeria, UK), most countries are statistically indistinguishable from the grand mean. Furthermore, point estimates of the effect of the broad treatment are nearly uniformly positive, and we can rule out large negative effects in most countries. Thus, we summarize the results as providing evidence that accurate normative information often increases intentions to accept COVID-19 vaccines with little risk of negative effects. We do not find sufficient evidence of international heterogeneity that would justify different guidance for different countries in this sample. The heterogeneity that is observed in country-level treatment effects could be partially explained by the variation in normative information shown to respondents, with countries with higher baseline vaccine acceptance associated with larger treatment effects (Supplementary Fig. S10). As a more explicit post hoc test of this, in Supplementary Fig. S11 we group the treatment into bins 20 percentage points wide and find providing higher normative information is associated with larger treatment effects (joint tests of equal treatment effects, $\chi^2(1) = 4.93, P = 0.03$ and $\chi^2(3) = 19.97, P < 0.001$, for the broad and narrow treatments, respectively).

In addition to the primary experiment embedded in the global survey[36], we conducted a supplementary survey in the United States over two waves to measure the link between vaccination intentions and self-reported vaccination uptake. This supplementary survey was much smaller scale ($n = 1350$), though we were able to explicitly follow-up with respondents with a first wave beginning April 2, 2021 and a follow-up wave beginning May 18, 2021. In this supplementary survey, we find that self-reported vaccination intentions are predictive of future, self-reported vaccination status (see Supplementary Note 8). If respondents in our international experiment were to be vaccinated at the same rate as those in this supplementary analysis, we would see a 23.1 percentage point increase in vaccination rates among those who were unsure but were induced to say they would probably accept a vaccine and a 17.2 percentage point increase in vaccination rates among those who would probably accept a vaccine but were induced to say they would definitely accept a vaccine.

## Robustness checks

An important limitation is that we are only able to estimate effects on intentions to accept a vaccine against COVID-19, which could differ from effects on vaccine uptake. While it has not been feasible to study interventions that measure take-up of the COVID-19 vaccine on a representative global population, we believe that the intervention studied here is less subject to various threats to validity—such as experimenter demand effects—that are typically a concern in survey experiments measuring intentions.

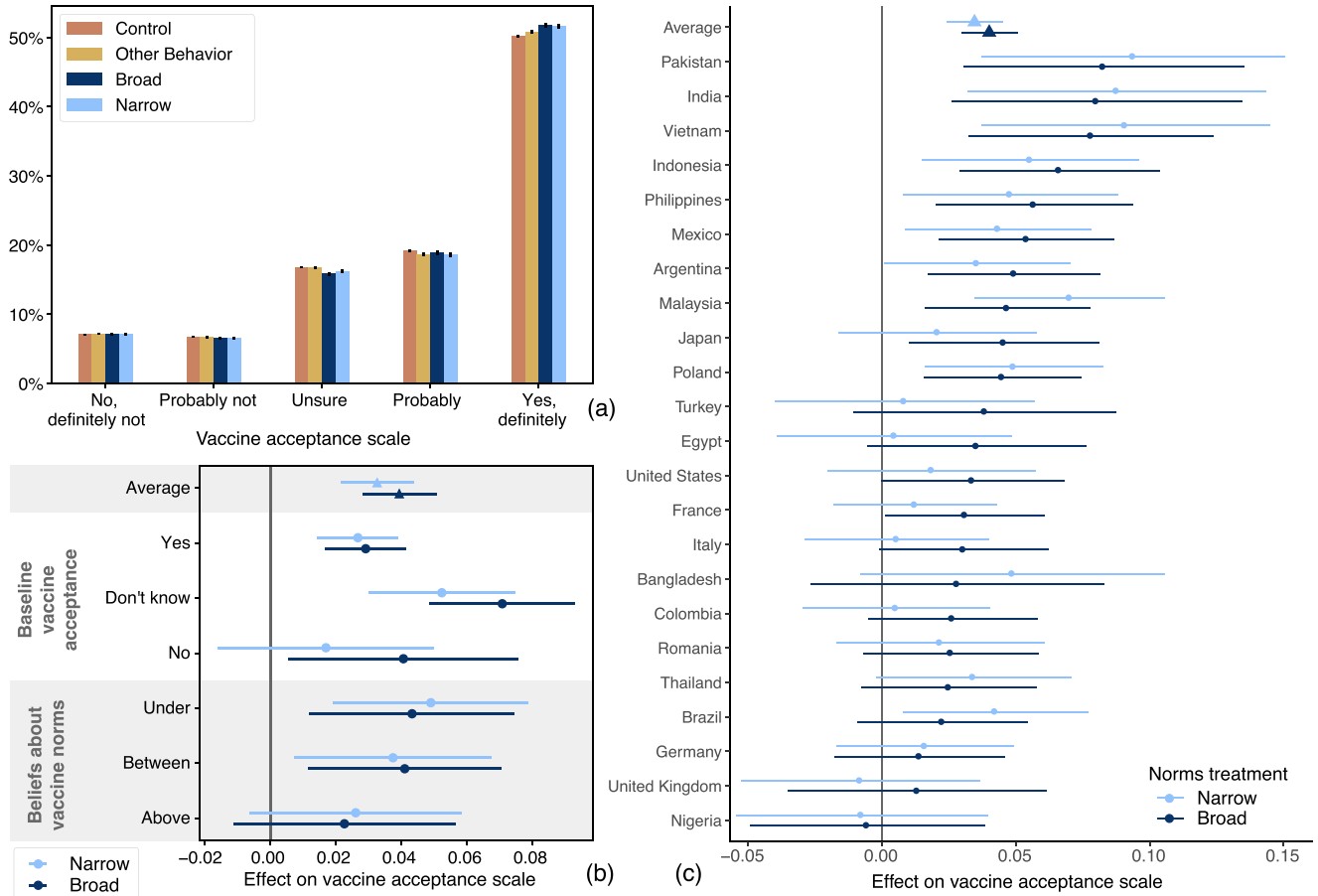

**Fig. 4 | Effect of intervention on vaccination intentions. a** The normative information treatments shift people to higher levels of vaccine acceptance, whether compared with receiving no information (control) or information about other, non-vaccine-acceptance norms (other behavior). The figure shows estimated distribution of vaccine acceptance responses for $n = 464,533$ respondents. **b** These estimated effects are largest for respondents who are uncertain about accepting a vaccine at baseline and respondents with baseline beliefs about descriptive norms that are under (rather than above or between) both of the levels of normative information provided in the treatments. There are $n = 365,593$ responses in the average analysis, $n = 362,438$ responses in the baseline vaccine acceptance analysis, and $n = 113,438$ responses in the beliefs about vaccine norms analysis. **c** While there is some country-level heterogeneity in these effects, point estimates of the effect of the broad normative information treatment are positive in all but one country ($n = 365,593$ responses). Error bars are 95% confidence intervals centered around mean estimates. Source data are provided as a Source Data file.

This randomized experiment was embedded in a survey with a more general advertised purpose that covers several topics, so normative information is not particularly prominent (Supplementary Note 1). In this broader survey, only 15% of questions were specific to vaccinations or social norms[36]. Furthermore, unlike other sampling frames with many sophisticated study participants (e.g., country-specific survey panels, Amazon Mechanical Turk), respondents are recruited from a broader population (Facebook users). In addition, we observe smaller effects for observable behaviors such as distancing and mask wearing, which would be surprising if researcher demand effects were driving the effects for vaccine acceptance.

A number of robustness checks increase our confidence that experimenter demand is not driving the result. As a first robustness check, we compare the outcome of respondents who receive the vaccine norm treatment to those receiving the treatment providing information about masks and distancing. The results are largely consistent and suggest that the vaccine information treatment increases vaccination intentions, while effects for distancing and masks are smaller and often not statistically distinguishable from zero. (Supplementary Fig. S6 and Supplementary Note 5). Moreover, we may expect researcher demand effects to be smaller when the information treatment and the outcome are not immediately adjacent. In all cases, for the vaccine acceptance outcome, there is always at least one

intervening screen of questions (the future mask wearing and distancing intentions questions). Furthermore, they are often separated by more than this. We consider a subset of respondents where the treatment and the outcome are separated by at least one "block" of questions between them. The results of this analysis are presented in Supplementary Fig. S12 and Supplementary Table S13 of Supplementary Note 5.1. The estimated effects of the vaccine treatments in this smaller sample are somewhat muted and less precise, but both significantly positive. Moreover, Supplementary Table S14 shows even with the larger gap between treatment and outcome, the information is still moving a relatively large share of people who are unsure or more negative to at least probably accepting the vaccine.

All analyses presented take advantage of survey weights that adjust the survey for sampling and non-response bias[37]. This is to make the analysis as representative as possible for the countries we survey. To motivate the use of weights, consider Supplementary Fig. S14a, which plots the estimated share of countries' population that is female. The unweighted estimates have substantial bias, and the weighted estimators reduce this bias. Formally, non-response weighting is justified by the assumption that data are missing at random (conditional on covariates used for weighting, respondents are a random sample of those sampled)[41]. While this is a strong assumption, we find it more plausible than the assumption required for an unweighted analysis that

assumes the sample is a random sample from the target population, which we can confidently reject (Supplementary Fig. S14a). As a robustness check, however, we run the analysis using unweighted estimators and find the treatment effects are robust to the use of weights (Supplementary Fig. S14b).

## Discussion

Framing vaccination as a social norm has been suggested as an effective approach to building COVID-19 vaccine confidence[42–44], but this recommendation has lacked direct evidence on a scalable messaging strategy using accurate information, which this international randomized experiment now contributes. Brewer et al.[16] document the case of a vaccine campaign by a major pharmacy retail chain in the United States that employed negative norms messaging to emphasize risks to individuals: "Get your flu shot today because 63% of your friends did not." Although such a strategy can reduce incentives to free-ride on vaccine herd immunity, its broader impact on social norm perceptions may render it ineffective. On the other hand, one might worry that accurate information about descriptive norms would simply feature pluralities or majorities that are too small to be effective. In general, the multimodal effects of descriptive norms on risk perceptions, pro-social motivations, and social conformity highlight the value of the evidence we provide here. In particular, our results across countries suggest that accurate normative information often increases intentions to accept COVID-19 vaccines, while generally ruling out large negative effects, and effects are largest in countries with higher norms. In addition, we find little evidence that providing normative information to those that overestimate vaccine acceptance results in decreased vaccination intentions. While our analysis finds some evidence that effects are smallest among those who overestimate the descriptive norm, the point estimates are positive (though statistically indistinguishable from zero), and we can rule out large negative effects. Taken together, this evidence suggests the positive effects from pro-social motivations and social conformity outweigh the possible negative effects from any free-riding on herd immunity. However, extrapolating the results of this experiment to much higher levels of the norm than presently observed for COVID-19 vaccine acceptance increases the theoretical likelihood that knowledge of the norm could trigger free-riding.

For social norms to be effective, it is critical that they are salient in the target population (e.g., wearing badges[45]). While in our randomized experiment norms are made salient through direct information treatments, the results have implications for communication to the public through health messaging campaigns and the news media. For example, if very high levels of vaccine uptake are needed to reach (even local) herd immunity[3] and to minimize severe illness[46], it is reasonable for news media to cover the challenges presented by vaccine hesitancy; but our results suggest that it is valuable to contextualize such reporting by repeatedly noting the widespread norm of accepting COVID-19 vaccines. Public health campaigns to increase acceptance of safe and effective vaccines can include information about descriptive norms. In an effort to influence the public, some public figures have documented receiving a COVID-19 vaccine in videos on television and social media. The positive effects of numeric summaries of everyday people's intentions documented here suggest that simple factual information about descriptive norms can similarly leverage social influence to increased vaccine acceptance. Pockets of negative attitudes toward vaccination put local communities at more risk, so emphasizing country-wide vaccination norms may prove critical for encouraging members of these communities to get vaccinated[3,47].

In addition to being salient, effective social norm interventions must be credible[48,49] and not inconsistent with strongly held beliefs[50]. This understanding helps explain a number of our findings. First, as mask wearing and physical distancing are easily observable behaviors in the community, any discrepancy in the descriptive norm provided

to individuals may be viewed skeptically, consistent with the smaller effects found for these preventative behaviors. Moreover, we observe the largest effects among those who are unsure if they will accept a vaccine consistent with the literature suggesting normative interventions are less effective when norms are inconsistent with beliefs[50].

How important are the effects of the factual descriptive normative messages studied here? Smaller-scale interventions that treated individuals with misinformation[51], pro-social messages[52], demographically tailored videos[53], text message reminders[54], or other informational content[55] have yielded similar or smaller effect sizes, while lacking the scalability and practical appeal of accurate descriptive norms. The positive effects of normative information about vaccine acceptance may reflect that people have little passive exposure to information about how many people in their communities and countries would accept a vaccine, or even have done so already. This result contrasts with other preventative behaviors (mask wearing and distancing), for which we observe smaller or no statistically significant effects (see Supplementary Note 5). Mask wearing and physical distancing are readily observable, require continued effort, and are ongoing activities (i.e., respondents have repeatedly chosen whether to perform them before). Vaccination decisions, however, are typically not easily observable to others, which could enhance the credibility of normative interventions about vaccines relative to observable behaviors[49]. Moreover, at the time of the study, vaccinations were not widely available to the public. This led to a substantial share of respondents being uncertain if they would accept a vaccine when offered one, and the treatment was most effective among these individuals. Individuals had repeatedly made decisions about behaviors such as mask wearing and physical distancing, suggesting there were fewer "fence-sitters" who are more likely to be influenced[40] and fatigue with such activities may have set in. We therefore think it is likely that as people make their own vaccination decisions and have more familiarity with social contacts and community members choosing to accept a vaccine, this type of normative information will become less impactful.

How will our results for intentions to accept vaccines translate into vaccine receipt? Prior studies exhibit important concordance between vaccination intentions and subsequent take-up[56]—and effects of treatments on each[57,58]. In addition, the intentions measured in the survey are predictive of the cross-country variation in vaccination shares (Supplementary Note 7). Moreover, the supplementary survey we fielded suggests that self-reported vaccination intentions are predictive of future vaccination status (Supplementary Note 8). While uncertainty remains in the extent to which the effects on intentions translate into actions, we can largely rule out negative effects from this information and the potential benefits appear to outweigh the relatively low costs of providing information. To what degree effects on intentions translate into increased vaccination depends on factors such as the ease of getting vaccinated. Thus, we encourage the use of these factual normative messages, as examined here; but we also emphasize the need for a range of interventions that lower real and perceived barriers to vaccination, remind people to get vaccinated[54], and leverage descriptive norms and social contagion more generally, such as in spreading information about how to obtain a vaccine[24]. Early trials combining multiple influence strategies and types of information, including descriptive social norms, have shown promise in this regard[59].

## Methods
### Consent

All participants were adults and consented to participation in the research via online forms. There were 484,239 participants in the experiment (44% female, modal age group 31–40). There were 1350 respondents who completed both the initial and follow-up supplemental survey (52% female, average age 40). Participants in the primary

study were not compensated; participants in the follow-up study were compensated through the online panel CloudResearch.

## Ethical approvals

The MIT Committee on the Use of Humans as Experimental Subjects approved the original survey (protocol E-2294), the randomized experiment (protocol E-2674), and the supplemental study (protocol E-3105) as exempt studies.

## Experiment overview

During an update to the survey on October 28, 2020, we introduced a prompt to all respondents that provided information about preventative behaviors in their country based on information from the survey. Although this information was provided to all respondents who completed the survey from an eligible country, the information was provided in a random order creating an experiment within the survey. For each eligible respondent, we showed the following message at a random position in the latter part of the survey:

> Your responses to this survey are helping researchers in your region and around the world understand how people are responding to COVID-19. For example, we estimate from survey responses in the previous month that [[country share]]% of people in your country say they [[broad or narrow]] [[preventative behavior]].

We filled in the blanks with one randomly chosen preventative behavior, a broad or narrow definition of the activity, and the true share of responses for the respondent's country. The three behaviors were vaccine acceptance, mask wearing, and social distancing. In the broad condition, we used a more inclusive definition of the preventative behavior, and the narrow condition used a more restrictive definition. For example, for vaccine acceptance, we either reported the share of people responding "Yes" or the share of people responding "Yes" or "Do not know" to the baseline vaccine acceptance question. The numbers are shown, which were updated with each wave, are displayed in Supplementary Fig. S3. We conduct a number of randomization and balance checks in Supplementary Note 3 (Supplementary Fig. S4), and the randomization appears to have occurred as expected.

Given the design of the survey intentionally ensured we are unable to identify any given survey respondent, we cannot rule out that some participants took the survey more than once, though the recruitment method was designed to not re-recruit participants within short periods. Given the size of our sample relative to the Facebook population, it is unlikely that this represents a substantial share of our responses.

We pre-registered our analysis plan, which we also updated to reflect continued data collection and our choice to eliminate the distancing information treatment in later waves. While we describe some of the main choices here, our pre-registered analysis plans can be viewed at https://osf.io/h2gwv/ and was initially submitted on October 28, 2020. The analysis of the experiment that is not described in the analysis plan is labeled post hoc (in particular, heterogeneity by baseline vaccine acceptance). In addition, the survey was initially expected to end in December 2020, but was extended until March 2021 and we use all the available data in all analyses. If we restrict to the original planned sample, the treatment effects of both the narrow and broad vaccine interventions are similar in magnitude and significant at the 1% level ($0.044 \pm 0.012$ s.e. and $0.055 \pm 0.013$ s.e. for narrow and broad treatments, respectively); the treatment effects on distancing and mask wearing are not statistically significant. After all data from the original period was collected, we modified the randomization to assign 2/3 of treated individuals to the vaccine treatment and 1/3 to the mask treatment. We removed the distancing treatment after collecting the pre-registered amount of data, as the question was less concrete and it had a non-statistically significant impact on beliefs (using other behaviors as a control group). We chose to emphasize vaccination in our analysis, after collecting and analyzing the full pre-registered sample size for all three preventative behaviors, because of the increasing policy relevance and imminent availability of vaccination to the public. Finally, one set of more complex analyses speculatively described in the analysis plan (hypothesis 3, "may suggest using instrumental variables analyses") has not been pursued. There are no other deviations from the pre-registered analysis plan.

## Data construction

Our dataset is constructed from the microdata described in ref. 36 using waves 9–19 of the survey (the randomized experiment began in wave 9). We use the variables collected in the survey that are described in Supplementary Note 2. We first code each outcome to a 5-point numerical scale. We then condition on being eligible for treatment and having a waves survey type (i.e., being in a country with continual data collection) to arrive at the full dataset of those eligible for treatment. Respondents in the snapshot survey may have received treatment if they self-reported being in a wave country; these individuals are removed as their weights will be for the wrong country. All randomization and balance checks described as "intent-to-treat" use this dataset. In our pre-registered analysis plan, we described how the sample would be restricted to those who completed the survey and for whom we received a full survey completion weight from Facebook. This removes ~40% of respondents, resulting in 484,239 respondents. For the main analysis comparing users who received the vaccine information treatment to control users (e.g., in Fig. 4b), there are 365,593 respondents.

## Experiment analysis

The results presented and elaborated on in the SI each use a similar pre-registered methodology that we briefly describe here. For the results in Figs. 3 and 4a, we estimate the following linear regression:

$$Y_i = \delta_0 + \sum_{j \in J} \delta_j D_i^j + \gamma X_i + \sum_{j \in J} \eta_j X_i D_i^j + \varepsilon_i, \tag{1}$$

where $Y_i$ is the outcome for individual $i$, $D_i^j$ is an indicator if individual $i$ received treatment $j \in J = \{\text{Broad, Narrow}\}$, and $X_i$ is a vector of centered covariates[60,61]. See section Supplementary Note 2 for the list of pre-registered covariates included in the analysis. All statistical inference uses heteroskedasticity-consistent Huber–White sandwich estimates of the variance–covariance matrix and all statistical tests are two-sided.

For heterogeneous treatment effects (Fig. 4b), we estimate a similar regression where covariates are centered at their subgroup-specific means. For brevity, we suppress the behavior index $k$ below.

$$Y_i = \sum_{b \in B} 1[b_i = b] \left( \delta_0^b + \sum_{j \in J} \delta_j^b D_{ij}^b + \gamma X_i + \sum_{j \in J} \eta_j^b X_i D_{ij}^b \right) + \varepsilon_i. \tag{2}$$

The analysis primarily used Python 3.8 with the following packages: numpy (1.21.2), pandas (1.3.0), patsy (0.5.1), scipy (1.6.2), stargazer (0.0.5), statsmodels (0.12.2).

## Mixed-effects model

In Fig. 4c, we report results from a linear mixed-effects model with coefficients that vary by country. This model is also described in our pre-registered analysis plan. Note that the coefficients for the overall (across-country) treatments effects in this model differ slightly from the estimates from the model in Eq. (1); that is, the "Average" points in Fig. 4b, c do not match exactly. As noted in our analysis plan, sandwich standard errors are not readily available here, so 95% confidence intervals are obtained by estimating the standard errors via a bootstrap. The mixed-effects modeling analysis was run using R version 3.5.1, and additional auxiliary analysis was run using R 4.0.21.

## Reporting summary

Further information on research design is available in the Nature Portfolio Reporting Summary linked to this article.

## Data availability

Documentation of the survey instrument and aggregated data from the survey are publicly available at https://covidsurvey.mit.edu. Researchers can request access to the raw (individual level) data from Facebook and MIT at https://dataforgood.fb.com/docs/preventive-health-survey-request-for-data-access/. Moreover, the aggregated data to recreate the figures of this paper have been deposited in https://github.com/alexmoehring/NormsIncreaseVaccineAcceptance[62] and are provided as Source Data with this paper. Source data are provided with this paper.

## Code availability

Analysis code to reproduce figures in the manuscript are available at https://github.com/alexmoehring/NormsIncreaseVaccineAcceptance[62].

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

## Acknowledgements

The COVID-19 Global Beliefs, Behaviors, and Norms Survey was a collaborative effort involving contributions from individuals at multiple institutions, especially the Massachusetts Institute of Technology, Johns Hopkins University, the Global Outbreak Alert and Response Network, the World Health Organization, and Facebook, with key contributions from Stella Babalola, Nina Gobat, Esther Kim, Kelsey Mulcahy, Praveen Raja, Stephanie Sasser, Dominick Shattuck, Jeni Stolow, Carlos Velasco, and Thomas Wynter. We thank Eytan Bakshy, Adam Berinsky, Daniel Björkegren, B.J. Fogg, Alex Leavitt, Solomon Messing, and David Rand. We thank the millions of respondents to this survey worldwide. This work was funded in part by a grant from Meta, which operates Facebook, to the MIT Initiative on the Digital Economy.

## Author contributions

A.M., A.C., K.G., M.R., S.A., and D.E. all contributed to the design of the survey and the randomized experiment. A.M. and D.E. wrote the pre-registered analysis plan. A.M. led the data analysis, with contributions from all authors. A.M. wrote the code to retrieve and merge the weights from Facebook; A.M., A.C., K.G., and D.E. accessed the resulting merged data. A.M., A.C., K.G., M.R., S.A., and D.E. all contributed to writing the paper.

## Competing interests

In addition to funding this work, Meta has sponsored a conference organized by S.A. and D.E.; M.A.R. serves on the advisory committee of a vaccine confidence fund created by Meta and Merck; D.E. was a consultant to Twitter during review and revision of this paper; A.C., S.A., and D.E. have received funding for other research from Meta. The remaining authors declare no competing interests.
