## [Peer Review File · Nature Communications]

Providing normative information increases intentions to
accept a COVID-19 vaccineEditorial Note: This manuscript has been previously reviewed at another journal that is not operating a transparent peer review scheme. This document only contains reviewer comments and rebuttal letters for versions considered at *Nature Communications*.

Reviewer #1 (Remarks to the Author):

I previously reviewed this manuscript at another journal. I have a very high opinion of this research, and I have the same view of this revised version. All of my suggestions have been addressed. I support acceptance of this paper. It would fit very well in a journal like Nature Communications, will be influential, and widely cited.

Reviewer #2 (Remarks to the Author):

This is a very well-executed paper. There is little to complain about the methods or analyses and I think that this paper can make a nice contribution to the literature, despite how crowded this field has become.

However, I'm quite unhappy with the very incomplete and (purposely?) selective citation of relevant literature. In particular, the authors' oversight to respect the many contributions by Cristina Bicchieri (I am not her) to this field are disturbing, especially since she -- and her collaborators -- have contributed to exactly this topic, both empirically and theoretically. It's particularly worrisome because the authors very obviously rely on her methodological framework: the idea that norms are a construct of one's empirical (what the authors call descriptive) and normative expectations with respect to one's reference group has already been spelled out theoretically in Bicchieri (2005) -- and more recently in Dimant & Bicchieri (2019) in the context of 'norm nudging', which is exactly what the authors are doing in their normative information provision intervention. The authors also fail to consider a rather relevant cross-cultural paper on norms and COVID-19 by Bicchieri et al. (2021) as well as Bicchieri et al. (2020) with respect to norm perceptions, norms change, and peer effects. If you read those papers closely you will also find a lot of relevant literature cited that does not involve Bicchieri.

I urge the authors to provide a much clearer account of the social norms framework that they are relying on to guide both their hypotheses and results.

It is important that we acknowledge the work of those who laid the foundation for future research and I encourage the authors to be much more thorough in this regard.

References

Bicchieri, C., 2005. The grammar of society: The nature and dynamics of social norms. Cambridge University Press.

Bicchieri, C. and Dimant, E., 2019. Nudging with care: The risks and benefits of social information. Public Choice, pp.1-22.

Bicchieri, C., Dimant, E., Gächter, S. and Nosenzo, D., 2020. Social proximity and the erosion of norm compliance. Preprint.

Bicchieri, C., Fatas, E., Aldama, A., Casas, A., Deshpande, I., Lauro, M., Parilli, C., Spohn, M., Pereira, P. and Wen, R., 2021. In science we (should) trust: Expectations and compliance across nine countries during the COVID-19 pandemic. PloS one, 16(6), p.e0252892.

Reviewer #3 (Remarks to the Author):

Vaccine hesitance, particularly in the midst of the covid-19 pandemic, is a major issue. The authors of this manuscript sought to evaluate whether providing information on descriptive norms (e.g. the % of the population who are vaccinated) to attempt to impact perceptions of these norms on intention to be vaccinated against covid-19. While a well-conducted study, the overall impression of this research was underwhelming. A small effect was detected, on the magnitude of a few percentage points, but this was a difference in intention to get vaccinated. Given the issues with the intention-to-action gap, it is unclear whether these findings would translate into substantial increases in covid-19 vaccine uptake.

Additionally, the focus on country-level data can be seen to be the most efficient way to present these data, as the data are more readily available than sub-national data, and requires less levels of tailoring for communications campaigns than sub-national data. However, given the heterogeneity in vaccine uptake (for example, the latest data from the US indicates 69% of the population has received 1+ covid vaccine dose and 59% are fully vaccinated, but with state-level fully vaccinated ranging from 46% to 72%). While the 59% value may resonate with citizens of a lower vaccinated state, it would likely not drive many people in a highly vaccinated state to be vaccinated.

The null findings related to mask wearing and distancing indicate that there may be more pull related to vaccine-related data presentation, but could also be indicative of "pandemic fatigue" given the longer period of time for which these NPI were recommended, whereas vaccination has only been available for <1 year. While it would be difficult to try to tease out these differences, the authors limitation around acute (e.g. vaccination) versus chronic (e.g. masking) actions seems incomplete.

The authors mention briefly some of the issues around perceptions of herd immunity and the free-rider issue. But, there is little more discussion of this, and given the point above about heterogeneity across the US (and likely for many other countries as well), this type of potential negative effect of the presentation of these descriptive norms should be considered more.

Reviewer Comments

Reviewer #1

I previously reviewed this manuscript at another journal. I have a very high opinion of this research, and I have the same view of this revised version. All of my suggestions have been addressed. I support acceptance of this paper. It would fit very well in a journal like Nature Communications, will be influential, and widely cited.

Thank you for your encouraging review of our manuscript. We are pleased that we have adequately addressed your suggestions.

Reviewer #2

This is a very well-executed paper. There is little to complain about the methods or analyses and I think that this paper can make a nice contribution to the literature, despite how crowded this field has become.

However, I'm quite unhappy with the very incomplete and (purposely?) selective citation of relevant literature. In particular, the authors' oversight to respect the many contributions by Cristina Bicchieri (I am not her) to this field are disturbing, especially since she -- and her collaborators -- have contributed to exactly this topic, both empirically and theoretically. It's particularly worrisome because the authors very obviously rely on her methodological framework: the idea that norms are a construct of one's empirical (what the authors call descriptive) and normative expectations with respect to one's reference group has already been spelled out theoretically in Bicchieri (2005) -- and more recently in Dimant & Bicchieri (2019) in the context of 'norm nudging', which is exactly what the authors are doing in their normative information provision intervention. The authors also fail to consider a rather relevant cross-cultural paper on norms and COVID-19 by Bicchieri et al. (2021) as well as Bicchieri et al. (2020) with respect to norm perceptions, norms change, and peer effects. If you read those papers closely you will also find a lot of relevant literature cited that does not involve Bicchieri.

I urge the authors to provide a much clearer account of the social norms framework that they are relying on to guide both their hypotheses and results.

It is important that we acknowledge the work of those who laid the foundation for future research and I encourage the authors to be much more thorough in this regard.

References

Bicchieri, C., 2005. *The grammar of society: The nature and dynamics of social norms*. Cambridge University Press.

Bicchieri, C. and Dimant, E., 2019. Nudging with care: The risks and benefits of social information. *Public Choice*, pp.1-22.

Bicchieri, C., Dimant, E., Gächter, S. and Nosenzo, D., 2020. Social proximity and the erosion of norm compliance. Preprint.

Bicchieri, C., Fatas, E., Aldama, A., Casas, A., Deshpande, I., Lauro, M., Parilli, C., Spohn, M., Pereira, P. and Wen, R., 2021. In science we (should) trust: Expectations and compliance across nine countries during the COVID-19 pandemic. *PloS one*, 16(6), p.e0252892.

Thank you for your detailed and encouraging review. We appreciate you highlighting gaps in our references to the existing literature. We have now read much of this work and have attempted to fill these gaps in the introduction and discussion section. In addition to adding citations to related work, we have expanded our discussion section to contextualize the results of the experiment within the theoretical framework of Dimant & Bicchieri (2019). We believe this revision has strengthened our discussion of the findings and hopefully more accurately portrays the prior literature our work builds on. To highlight the new discussion, below are two excerpts where we try to interpret our results in light of our improved understanding of the existing literature. In addition to this discussion we also added several citations in the literature review portion. Please see the comparison file for all changes.

“In addition to being salient, effective social norm interventions must be credible (Stibe et al, 2016, Bicchieri and Dimant, 2019) and not inconsistent with strongly held beliefs (Bicchieri and Mercier, 2014). This understanding helps explain a number of our findings. First, as mask wearing and physical distancing are easily observable behaviors in the community, any discrepancy in the descriptive norm provided to individuals may be viewed skeptically, consistent with the smaller effects found for these preventative behaviors. Moreover, we observe the largest effects among those who are unsure if they will accept a vaccine consistent with the literature suggesting normative interventions are less effective when norms are inconsistent with beliefs (Bicchieri and Mercier, 2014).”

“Masking and physical distancing are readily observable, require continued effort, and are ongoing activities (i.e., respondents have repeatedly chosen whether to perform them before) and readily observable in public. Vaccination decisions, however, are typically not easily observable to others which could enhance the credibility of normative interventions about vaccines relative to observable behaviors (Bicchieri and Dimant, 2019).”

Reviewer #3

Vaccine hesitance, particularly in the midst of the covid-19 pandemic, is a major issue. The authors of this manuscript sought to evaluate whether providing information on descriptive norms (e.g. the % of the population who are vaccinated) to attempt to impact perceptions of these norms on intention to be vaccinated against covid-19. While a well-conducted study, the overall impression of this research was underwhelming. A small effect was detected, on the magnitude of a few percentage points, but this was a difference in intention to get vaccinated. Given the issues with the intention-to-action gap, it is unclear whether these findings would translate into substantial increases in covid-19 vaccine uptake.

Thank you for the careful read of our manuscript and the thoughtful suggestions. We do not want to overstate the magnitude of the effect found in our study and have tempered the language describing the size of the effects found. We appreciate the criticism that intentions may not translate into actions. We try to provide evidence in the supplementary materials that the intentions measured in the survey are predictive of cross country variation in actual vaccine uptake (Section S8) and that self-reported intentions predict self-reported behaviors (Section S9). Of course, this does not fully resolve your concern and we attempt to appropriately acknowledge this and caveat our results in the manuscript.

Additionally, the focus on country-level data can be seen to be the most efficient way to present these data, as the data are more readily available than sub-national data, and requires less levels of tailoring for communications campaigns than sub-national data. However, given the heterogeneity in vaccine uptake (for example, the latest data from the US indicates 69% of the population has received 1+ covid vaccine dose and 59% are fully vaccinated, but with state-level fully vaccinated ranging from 46% to 72%. While the 59% value may resonate with citizens of a lower vaccinated state, it would likely not drive many people in a highly vaccinated state to be vaccinated.

We agree that within an individual country there may be substantial heterogeneity across regions in the effectiveness of such an intervention. In particular, it is a valid concern that regions with high vaccine acceptance may not be swayed (or even experience negative effects) to learning the average descriptive norm. In our data, we do find some evidence that treatment effects are smallest among those who overestimate the descriptive norm. That is, for those who believe more individuals in their region will receive a vaccine than the number we show them. That said, the point estimates are positive for this group,

though statistically indistinguishable from zero. We have revised the discussion section to try and address this point of concern.

“In particular, our results across countries suggest that accurate normative information often increases intentions to accept COVID-19 vaccines, while generally ruling out large negative effects and effects are largest in countries with higher norms. In addition, we find little evidence that providing normative information to those that overestimate vaccine acceptance results in decreased vaccination intentions. While our analysis finds some evidence that effects are smallest among those who overestimate the descriptive norm, the point estimates are positive (though statistically indistinguishable from zero) and we can rule out large negative effects. Taken together, this evidence suggests the positive effects from pro-social motivations and social conformity outweigh the possible negative effects from free-riding on herd immunity. Though, extrapolating the results of this experiment to much higher levels of the norm than presently observed for COVID-19 vaccine acceptance increases the likelihood that knowledge of the norm could trigger free-riding.”

The null findings related to mask wearing and distancing indicate that there may be more pull related to vaccine-related data presentation, but could also be indicative of "pandemic fatigue" given the longer period of time for which these NPI were recommended, whereas vaccination has only been available for <1 year. While it would be difficult to try to tease out these differences, the authors limitation around acute (e.g. vaccination) versus chronic (e.g. masking) actions seems incomplete.

We appreciate the alternative interpretation and agree that normative information regarding vaccinations was likely more effective in our experiment relative to masking and physical distancing due to vaccinations being new, while NPIs had been recommended for quite some time. We have tried to add additional nuance to this discussion to address this concern, and appropriately caveat the results. To do so, we highlight that as individuals begin to make decisions it is likely that the effectiveness of the normative intervention may wane.

“Masking and physical distancing are readily observable, require continued effort, and are ongoing activities (i.e., respondents have repeatedly chosen whether to perform them before) and readily observable in public. Vaccination decisions, however, are typically not easily observable to others which could enhance the credibility of normative interventions about vaccines relative to observable behaviors (Bicchieri and Dimant, 2019). Moreover, at the time of the study vaccinations were not widely available to the public. This led to a substantial share of respondents being uncertain if they would accept a vaccine when offered one, and the treatment was most effective among these individuals. Individuals had repeatedly made decisions about behaviors such as masking and physical distancing, suggesting there were fewer “fence-sitters” who are more

likely to be influenced (Betsch et al, 2015) and fatigue with such activities may have set in. We therefore think it is likely that as people make their own vaccination decisions and have more familiarity with social contacts and community members choosing to accept a vaccine, this type of normative information will become less impactful.”

The authors mention briefly some of the issues around perceptions of herd immunity and the free-rider issue. But, there is little more discussion of this, and given the point above about heterogeneity across the US (and likely for many other countries as well), this type of potential negative effect of the presentation of these descriptive norms should be considered more.

Thank you for the suggestion. We have added additional discussion to emphasize that our analysis could generally rule out large negative effects that would be consistent with free-riding. Moreover, while our data do not allow us to precisely study heterogeneity within a country, across countries we find that countries with larger descriptive norms typically have larger treatment effects. (Of course, at much higher levels of the norm than presently observed for COVID-19 vaccine acceptance, perhaps increased knowledge of the norm could trigger free-riding.) This discussion is included in the excerpts copied above as well as the sentence below when discussing the practical value of the intervention.

“While uncertainty remains in the extent to which the effects on intentions translate into actions, we can largely rule out negative effects from this information and the potential benefits appear to outweigh the relatively low costs of providing information.”

Reviewer #2 (Remarks to the Author):

I am satisfied with the revision in that all my comments have been addressed adequately.

Reviewer #3 (Remarks to the Author):

All prior comments have been sufficiently addressed.